

# *tran*-SAS v1.0: a numerical model to compute catchment-scale hydrologic *tran*sport using StorAge Selection functions

Paolo Benettin[1] and Enrico Bertuzzo[2]

[1]Laboratory of Ecohydrology ENAC/IIE/ECHO, École Polytechinque Fédérale de Lausanne (EPFL), Lausanne, Switzerland.
[2]Department of Environmental Sciences, Informatics and Statistics, Ca' Foscari University of Venice, Venice, Italy

*Correspondence to:* Paolo Benettin (paolo.benettin@epfl.ch)

**Abstract.** This paper presents the 'tran-SAS' package, which includes a set of codes to model solute transport and water residence times through a hydrological system. The model is based on a catchment-scale approach that aims at reproducing the integrated response of the system at one of its outlets. The codes are implemented in MATLAB and are meant to be easy to edit, so that users with minimal programming knowledge can adapt them to the desired application. The problem of large-scale solute transport has both theoretical and practical implications. On one side, the ability to represent the ensemble of water flow trajectories through a heterogeneous system helps unraveling streamflow generation processes and allows making inferences on plant-water interactions. On the other side, transport models are a practical tool that can be used to estimate the persistence of solutes in the environment. The core of the package is based on the implementation of an age Master Equation (ME), which is solved using general StorAge Selection (SAS) functions. The age ME is first converted into a set of ordinary differential equations, each addressing the transport of an individual precipitation input through the catchment, and then it is discretized using an explicit numerical scheme. Results show that the implementation is efficient and allows the model to run in short times. The numerical accuracy is critically evaluated and it is shown to be satisfactory in most cases of hydrologic interest. Additionally, a higher-order implementation is provided within the package to evaluate and, if necessary, to improve the numerical accuracy of the results. The codes can be used to model streamflow age and solute concentration, but a number of additional outputs can be obtained by editing the codes to further advance the ability to understand and model catchment transport processes.

## 1 Introduction

The field of hydrologic transport focuses on how water flows through a watershed and mobilizes solutes towards the catchment outlets. The proper representation of transport processes is important for a number of purposes such as understanding streamflow generation processes (Weiler et al., 2003; McGuire and McDonnell, 2010; McMillan et al., 2012), modeling the fate of nutrients and pollutants (Jackson et al., 2007; Hrachowitz et al., 2015), characterizing how watersheds respond to change (Kauffman et al., 2003; Oda et al., 2009; Danesh-Yazdi et al., 2016; Wilusz et al., 2017) and estimating solute mass export to stream (Destouni et al., 2010; Maher, 2011). The spatiotemporal evolution of a solute is typically expressed (Rinaldo and



Marani, 1987; Hrachowitz et al., 2016) as a combination of displacements, due to the carrier motion, and biogeochemical reactions, due to the interactions with the surrounding environment.

Water trajectories within a catchment are usually considered from the time water enters as precipitation to the time it leaves as discharge or evapotranspiration. As watersheds are heterogeneous and subject to time-variant atmospheric forcing, water flowpaths have marked spatiotemporal variability. For this reason, a formulation of transport by travel time distributions (see Cvetkovic and Dagan, 1994; Botter et al., 2005) can be particularly convenient as it allows transforming complex 3D trajectories into a single variable: the travel time, i.e. the time elapsed from the entrance of a water particle to its exit.

While early catchment-scale approaches (see McGuire and McDonnell, 2006) focused on the identification of an appropriate shape for the travel time distributions (TTD), emphasis has recently been put on a new generation of catchment-scale transport models, where TTDs result from a mass balance equation rather than being assigned *a priori* (Botter et al., 2011). As a consequence, TTDs change through time, as observed experimentally (e.g. Queloz et al., 2015a; Kim et al., 2016) and as required for consistency with mass conservation. This approach has the advantage of being consistent with the observed hydrologic fluxes and follows from the formulation of an age Master Equation (ME) (Botter et al., 2011), describing the age-time evolution of each individual precipitation input after entering the catchment. The key ingredient of this new approach is the "StorAge Selection" (SAS) function, which describes how storage volumes of different ages contribute to discharge (and evapotranspiration) fluxes. The direct use of SAS functions has already provided insights on water age in headwater catchments (van der Velde et al., 2012, 2015; Harman, 2015; Benettin et al., 2017; Wilusz et al., 2017), intensively managed landscapes (Danesh-Yazdi et al., 2016), lysimeter experiments (Queloz et al., 2015b; Kim et al., 2016), reach-scale hyporheic transport (Harman et al., 2016), and it has also been applied to non-hydrologic systems like bird migrations (Drever and Hrachowitz, 2017). In principle, applications can be extended to any system where the chronology of the inputs plays a role in the output composition.

The new theoretical formulation has improved capabilities, but the numerical implementation of the governing equations is more demanding than in traditional methods like the lumped convolution approach (e.g. Maloszewski and Zuber, 1993). This can represent a barrier to the diffusion of the new models, preventing their widespread use in transport processes investigation. To make the use of the new theory more accessible, the *tran*-SAS package includes a basic numerical model that solves the age ME using arbitrary SAS functions. The model is developed to simulate the transport of tracers in watershed systems, but it can be extended to other hydrologic systems (e.g. water circulation in lakes and oceans). The numerical code is written in MATLAB and it is intended to be intuitive and easy to edit, hence minimal programming knowledge should be sufficient to adapt it to the desired application.

The specific objectives of this paper are: i) provide a numerical model that solves the Age Master Equation with any form of the SAS functions in a computationally efficient way, ii) show the potential of the model for simulating catchment-scale solute transport, and iii) assess the numerical accuracy of the model for different aggregation timesteps.



## 2   Model Description

The model implemented in *tran*-SAS solves the age ME by means of general SAS functions and uses the solution to compute the concentration of an ideal tracer (conservative and passive to vegetation uptake) in streamflow. The model is described here using hydrologic terminology and applications.

### 2.1   Definitions

The theoretical framework relies on the works by Botter et al. (2011); van der Velde et al. (2012); Harman (2015); Benettin et al. (2015b) and requires knowledge of the input/output water fluxes to/from the catchment and the initial water storage. Moreover, to compute the evolution of solute concentration in the storage and in the out-fluxes, the input solute concentration must be known. We consider a typical hydrologic system with precipitation $J(t)$ as input and evapotranspiration $ET(t)$ and streamflow $Q(t)$ as outputs. The total system storage is obtained as $S(t) = S_0 + V(t)$ where $S_0$ is the initial storage in the system and $V(t)$ are the storage variations obtained from the hydrologic balance equation $dV/dt = J - ET - Q$. Tracer concentration in precipitation is indicated as $C_J(t)$.

The system state variable is the age distribution of the water storage. Indeed, at any time $t$, the water storage is comprised of precipitation inputs that occurred in the past and that have not left the system yet. Each of these past inputs can be associated with an age $T$, representing the time elapsed since its entrance into the watershed. Hence, at any time $t$ the storage is characterized by a distribution of ages $p_S(T,t)$. Similarly, discharge and evapotranspiration fluxes are characterized by age distributions $p_Q(T,t)$ and $p_{ET}(T,t)$, respectively. Each water parcel in storage can also be characterized by its solute concentration $C_S(T,t)$, which in case of an ideal tracer is equal to the concentration of precipitation upon entering the catchment $C_J(t-T)$. A useful, transformed version of the storage age distribution is the rank storage $S_T$ which is defined as $S_T(T,t) = S(t) \int_0^T p_S(\tau,t)d\tau$ and represents the volume in storage younger than $T$ at time $t$.

The key element of the formulation is the SAS function, which formalizes the functional relationship between the age distribution of the system storage and that of the outflows. Different forms have been proposed to express the SAS function directly as a function of age or as a derived distribution of the storage age distribution, (e.g. *absolute*, *fractional* or *ranked* SAS functions, see Harman (2015)). For numerical convenience, SAS functions are here expressed as cumulative probability distributions (CDF) of the rank storage, for both discharge ($\Omega_Q(S_T,t)$) and evapotranspiration ($\Omega_{ET}(S_T,t)$). Namely, $\Omega_Q(S_T,t)$ is, at any time $t$, the fraction of total discharge which is produced by $S_T(T,t)$. Hence, it is equal to the fraction of discharge younger than $T$. The corresponding probability density functions are indicated as $\omega_Q(S_T,t)$ and $\omega_{ET}(S_T,t)$. Main model variables are illustrated in Figure 1.

### 2.2   The Age Master Equation

The age ME (Botter et al., 2011) can be seen as a hydrologic balance applied to every parcel of water stored in the catchment. Two different equations can be formulated, that describe the forward-in-time or the backward-in-time process (Benettin et al., 2015b; Calabrese and Porporato, 2015; Rigon et al., 2016). Here, we focus on the backward form, as it is the most convenient





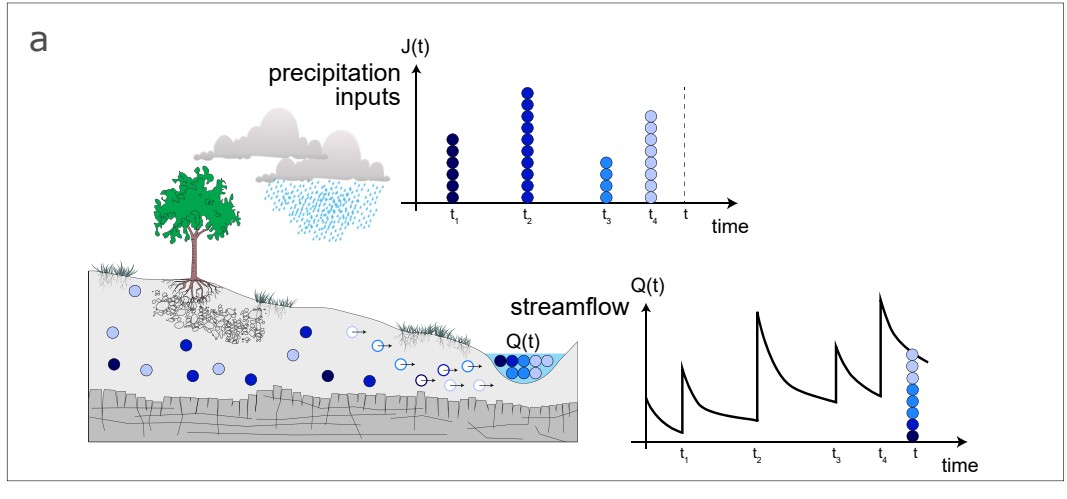

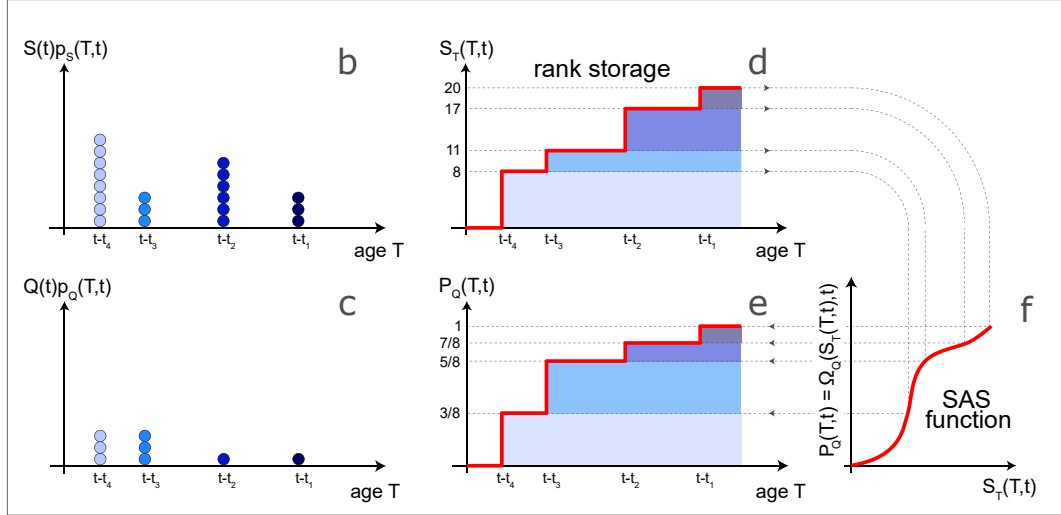

**Figure 1.** Conceptual illustration of the main variables of the theoretical formulation. Precipitation volumes are represented through coloured circles, with darker colours indicating the older precipitations with respect to current time $t$. Due to transport and mixing processes, precipitation volumes are retained in the catchment storage and released to streamflow (plot a). Both the catchment storage and its outfluxes are characterized by a distribution of ages (plots b and c). For example, the youngest water (age $t - t_4$, light blue colour) accounts for 8/20 of the storage and 3/8 of streamflow. By cumulating such distributions one gets the rank storage $S_T(T,t)$ and the cumulative discharge age distribution $P_Q(T,t)$ (plots d and e, red lines). The relationship between $S_T(T,t)$ and $P_Q(T,t)$ is quantified by the SAS function $\Omega_Q(S_T,t)$ (plot f).

to model solute concentration in streamflow. The backward form of the ME can be written in a number of equivalent forms that have been proposed in the literature (e.g. Botter et al., 2011; van der Velde et al., 2012; Harman, 2015). Here, we employ the cumulative version, which has a less intuitive physical interpretation but a better suitability to numerical implementation.





The complete set of equations reads:

$$\frac{\partial S_T(T,t)}{\partial t} + \frac{\partial S_T(T,t)}{\partial T} = J(t) - Q(t)\,\Omega_Q(S_T(T,t),t) - ET(t)\,\Omega_{ET}(S_T(T,t),t)\,, \tag{1}$$

Initial Condition: $S_T(T, t=0) = S_{T_0}\,,$ (2)

Boundary Condition: $S_T(T=0, t) = 0\,,$ (3)

where the initial condition $S_{T_0}$ indicates some initial distribution of the rank storage at time 0. Note that to ensure that $p_S$, $p_Q$ and $p_{ET}$ are distributions over the age domain $(0, +\infty)$, the SAS functions must verify the condition $\Omega_Q(S_T \to S(t), t) = \Omega_{ET}(S_T \to S(t), t) = 1$. This condition, however, is automatically verified as the SAS functions were defined as CDFs.

The solution of equation (1) gives the rank storage $S_T(T,t)$, from which the discharge age distributions $p_Q(T,t)$ can be obtained as:

$$p_Q(T,t) = \frac{\partial P_Q(T,t)}{\partial T} = \frac{\partial \Omega_Q(S_T(T,t),t)}{\partial T} = \frac{\partial \Omega_Q(S_T,t)}{\partial S_T}\frac{\partial S_T}{\partial T}\,, \tag{4}$$

where $P_Q(T,t)$ is the cumulative distribution of $p_Q(T,t)$ and $P_Q(T,t) = \Omega_Q(S_T,t)$ by definition. Stream solute concentration $C_Q(t)$ follows from:

$$C_Q(t) = \int_0^\infty C_S(T,t)p_Q(T,t)dT\,. \tag{5}$$

The same reasoning applies to the age distributions and concentration of the evapotranspiration flux.

## 2.3 The SAS functions

As explained in section 2.1, SAS functions are CDF's over the finite interval $[0, S(t)]$. A simple class of probability distributions that is suitable to serve as SAS function is the power-law distribution (Queloz et al., 2015b; Benettin et al., 2017), which takes the form:

$$\Omega(S_T,t) = \left[\frac{S_T(T,t)}{S(t)}\right]^{\mathbf{k}} = \left[\frac{S_T(T,t)}{\mathbf{S_0} + V(t)}\right]^{\mathbf{k}} \tag{6}$$

The parameter $\mathbf{k} \in (0, +\infty)$ controls the affinity of the outflow for relatively younger/older water in storage. Specifically, $k < 1$ [$k > 1$] implies affinity for young [old] water, whereas the case $k = 1$ represents "random sampling", i.e. outfluxes select water irrespective of its age. $\mathbf{k}$ can be conveniently made time-variant (e.g. dependent on the system wetness) to account for possible changes in the properties of the system (see van der Velde et al., 2015; Harman, 2015). Equation (6) also requires knowledge of the initial storage in the system $\mathbf{S_0}$, which can be difficult to estimate experimentally and it is often treated as a calibration parameter. When using power-law SAS functions for both $Q$ and $ET$, the system only requires 3 calibration parameters: $k_Q$, $k_{ET}$ and $S_0$. Different classes of probability distributions can be used to have more flexibility in the SAS function shape, e.g. the beta (van der Velde et al., 2012; Drever and Hrachowitz, 2017) or the Gamma (Harman, 2015; Wilusz et al., 2017) distributions. Such functions can be more difficult to implement numerically, but they are usually available in software libraries.



## 2.4 The special case of well-mixed/random-sampling

In case all the outflows remove the stored ages proportionally to their abundance, the outflow age distributions become a perfect sample (or *random* sample) of the storage age distribution. The SAS functions in this case assume the linear form $\Omega_Q(S_T,t) = \Omega_{ET}(S_T,t) = S_T(T,t)/S(t)$ and equation (1) has analytical solution (Botter, 2012):

$$p_S(T,t) = p_Q(T,t) = \frac{J(t-T)}{S(t)} \exp\left[ -\int_{t-T}^{t} \frac{Q(\tau) + ET(\tau)}{S(\tau)} d\tau \right] \tag{7}$$

Equation (7) can be seen as a generalization of the linear reservoir equation to fluctuating storage. Indeed, in the special case of a stationary system, where $J = Q + ET$ and the ratio $J/S$ is a constant $c$, equation (7) takes the simple form $p_S(T) = c \exp(-cT)$.

## 3 Model Implementation

### 3.1 Problem Discretization

Equation (1) does not have exact solution, except for the particular case of randomly sampled storage (section 2.4), so in general a numerical implementation is required. Following the approach by Queloz et al. (2015b) and Harman (2015), the partial differential equation (1) is first converted into a set of ordinary differential equations using the method of characteristics. Indeed, along a characteristic line of the type $t = T + t_0$, equation (1) simplifies into an ordinary differential equation in the single variable $T$:

$$\frac{dS_T(T,T+t_0)}{dT} = J(T+t_0) - Q(T+t_0)\,\Omega_Q(S_T,T+t_0) - ET(T+t_0)\,\Omega_{ET}(S_T,T+t_0), \tag{8}$$

with initial conditions $S_T(0,t_0) = 0$. In this context, reformulating the problem along characteristic lines means following the variable $S_T(T,T+t_0)$, i.e. the fraction of storage younger than the water input entered in $t_0$. The solution $S_T(T,T+t_0)$ starts from the value 0, corresponding to the time $t_0$ when the input enters. Then, as time (and age) grows, $S_T(T,T+t_0)$ increases when precipitation $J$ introduces younger water into the system and decreases when out-fluxes $Q$ and $ET$ withdraw water younger than $T$. The solution eventually reaches (asymptotically) the total storage in the system, as all the water is younger than the input that had entered in $t_0$.

We discretize time and age using the same time steps $\Delta T = \Delta t = h$, resulting in $T_i = i \cdot h$ and $t_j = j \cdot h$, with $i,j \in \mathbb{N}$ and we use the convention that the discrete variables $T_i$ and $t_j$ refer to the beginning of the timestep. To simplify the notation, square brackets are used to indicate the numerical evaluation of a function and the indexes $i$ and $j$ are used for $T_i$ and $t_j$ respectively. For example, $f[i,j]$ indicates the numerical evaluation of function $f(T_i,t_j)$. The conventions used for the discretization are illustrated in Figure 2. For numerical convenience and because real-world data often represent an average over a certain time-interval, all fluxes ($J$, $Q$, $ET$) are considered as averages over the time step $h$ (e.g., $J[j] = 1/h \int_{jh}^{(j+1)h} J(\tau)d\tau$). As a consequence, storage variations obtained from a hydrologic balance are linear during a timestep and each value refers to the beginning of the timestep.





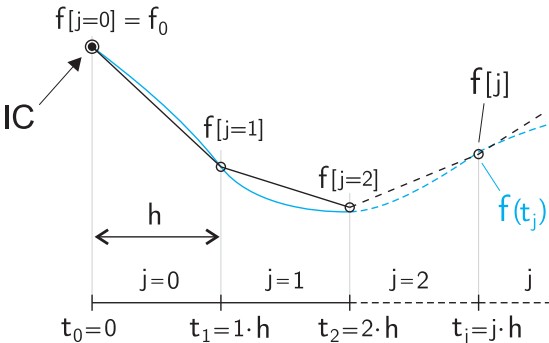

**Figure 2.** Illustration of the conventions used to discretize the time domain. Time steps have a fixed length $h$ (e.g. 12 hours) and each time step $j$ starts in $t_j = j \cdot h$. The numerical evaluation of a function $f$ at time $t_j$ is indicated as $f[j]$.

To solve equation (8), we implement a forward Euler scheme. This explicit numerical scheme is intuitive and fast to solve, and its numerical accuracy is shown to be satisfactory for many hydrologic applications (see model verification, section 5.1). By terming $\Omega[i,j] = \Omega(S_T[i,j], t_j)$, the discretized problem becomes:

$$S_T[i+1, j+1] = S_T[i,j] + h \cdot (J[j] - Q[j]\,\Omega_Q[i,j] - ET[j]\,\Omega_{ET}[i,j]) \tag{9}$$

for $i,j \in [0,N]$, with $N$ indicating the number of timesteps in the simulation, and boundary condition $S_T[0,j] = 0$. In a pure forward Euler scheme, this boundary condition implies that $\Omega[0,j] = \Omega(0, t_j) = 0$, meaning that no input can be part of an output during the same timestep. This can be a limitation for catchment applications, where "event" water is often not negligible and it can bear important information on catchment form and function. For this reason, in equation (9) we use a modified $\Omega^*$ defined as:

$$\Omega^*[i,j] = \Omega(S_T[i,j] + e[j], t_j) \tag{10}$$

where $e[j]$ is an estimate of the youngest water stored in the system at the end of time step $j$. Such an estimate is here obtained as $e[j] = \max(0, J[j] - Q[j]\,\Omega_Q[1, j-1] - ET[j]\,\Omega_{ET}[1, j-1]$, i.e. it is a water balance for current precipitation input using the SAS functions evaluated at previous timestep. The classic Euler scheme is returned if $e[j] = 0$. This modification of the classic numerical scheme only affects the behavior of the youngest age in the system and it is a simple and efficient way to

account for transport of event water. The accuracy of this numerical scheme is evaluated in Section 5.1.

## 3.2 Numerical routine

The model solves equation (9) by implementing an external for-loop on $j$ (i.e. on the chronologic time) and an internal for-loop on $i$ (i.e. on the ages). This means that during one timestep $j$ all the characteristic curves (equation (9)) are updated by one timestep. The internal loop is implemented using vector operations. The vector length is indicated as $n_j$ and it depends on the

number of age classes (which is also the number of characteristic curves) that are included in the computations at time $j$ (see section 3.3). At any time step, the two fundamental operations to solve the discretized ME are:





- compute $\Omega_Q^*[i,j]$ and $\Omega_{ET}^*[i,j]$ using equation (10);

- compute $S_T[i,j]$ using equation (9) for $i \in [1, n_j]$;

To compute the model output, further operations are required. In particular:

- update $C_S[i,j] = C_J[i-j]$, valid for conservative solutes entering through precipitation

- compute $p_Q[i,j] \cdot h = \Omega_Q[i,j] - \Omega_Q[i-1,j]$;

- compute $C_Q[j] = \sum_{i=1}^{n_j} C_S[i,j] \cdot p_Q[i,j] \cdot h$;

Starting from these basic routines, many additional operations can be implemented, to e.g. characterize the non-conservative behavior of solutes or to compute some age distribution statistics.

## 3.3 Additional numerical details

10 A first issue that the model needs to take into account is that age distributions are defined over an age domain $[0, +\infty)$, meaning that the rank storage is made of an infinite number of elements where the oldest elements typically represent infinitesimal stored volumes. To have a finite number of elements in the computations, an arbitrary old fraction of rank storage can be considered as a single undifferentiated volume of "older" water. This allows merging a high number of very little residual volumes into a single "old" pool. Such pool is here defined as the volume $S_T(T,t) > S_{th}$, where $S_{th}$ is a numerical parameter that can be 15 fixed for each different application. $S_{th}$ also defines the age $T_{th}$, corresponding to $S(T = T_{th}, t) = S_{th}$, which indicates the oldest age that is computed individually. Numerically, the parameter $S_{th}$ controls the number $n_j$ of age classes (or equivalently rank storage volumes) that are taken into account in the computations. $S_{th}$ should be chosen so that the number of elements used in the computations remains small but the numerical accuracy is not compromised. It can be convenient to define a non-dimensional threshold $f_{th} \in [0,1]$ such that $S_{th} = f_{th} S(t)$. In this case, a value $f_{th} = 0.9$ means that the old pool comprises 20 the oldest 10% of the water storage. When $f_{th} = 1$ no old pool is taken into account. Once a storage element is merged to the old pool, its individual age and concentration properties cannot be retrieved, but the mean properties of the old pool like the mean solute concentration are preserved.

A second, connected problem regards the initial conditions of the system, i.e. the unknown storage age distribution to be used at the beginning of the calculations. In the absence of information, the initial storage can be considered as one single 25 old pool, hence the initial number of age classes $n_0$ is equal to 1. Once computations start, new elements are introduced and accounted for in the balance, reducing the impact and the influence of the initial conditions. The old pool gets progressively smaller (and vector length $n_j$ larger) until it reaches the stationary value defined by $S_{th}$. An initial spinup period can be used to initialize the ME balance and reduce the size of the initial old water pool. The influence of the initial conditions decreases with time, but given the long timescales that characterize transport processes, it is never completely exhausted. This has little impact 30 on the output concentration but it limits the maximum computable age to the time elapsed since the start of the simulation.

The computational time of a simulation can be reduced by not accounting for zero-precipitation inputs as they have no influence in the balance but increase the number of operations required at each time step. In such a case, however, the position





**Table 1.** Description of the discharge SAS functions used in the application. All the functions were tested with the same initial total storage $S_0$=1000 mm.

| name | type | parameters | value |
|------|------|-----------|-------|
| $\omega_1$ | power law time variant | $k_{Q1}$ | 0.3 |
|  |  | $k_{Q2}$ | 0.9 |
| $\omega_2$ | power law | $k_Q$ | 0.7 |
| $\omega_3$ | random sampling | - | - |
| $\omega_4$ | beta | $a$ | 1.5 |
|  |  | $b$ | 0.8 |

of an element in the vector does not correspond with its age anymore and age has to be counted separately. To keep the model intuitive, we decided to not remove zero-precipitation inputs.

## 4 Application Example

The code comes with example virtual data that can be used to evaluate the model capabilities. Four years of hydrologic data
were obtained from recorded precipitation and streamflow at the Mebre-Aval station near Lausanne (CH). Evapotranspiration was obtained from regional daily estimates around the Lausanne area and modified to match the long-term mass balance. On average, yearly precipitation is 1100 mm, discharge is 580 mm (53% of precipitation) and evapotranspiration is 520 mm. The storage variations, computed by solving the hydrologic balance, were normalized to the interval [0,1] to serve as a non-dimensional metric of catchment wetness (variable $wi$). Overall, the data are not meant to be representative of a particular
location, but they constitute a realistic set of hydrologic variables to test the model.

The code was run on the example data using the 4 illustrative shapes for the discharge SAS function listed in Table 1. All simulations share the following settings: 12-h timestep, 4-year spinup period obtained by repeating the example data, storage threshold $f_{th}$=1 (i.e., no old-pool schematization), initial storage parameter $S_0$=1000, evapotranspiration SAS function selected as a power law with parameter $k$=1 (equivalent to a random sampling). The different shapes for the discharge SAS function
were selected to test different functional forms (power law, power law time variant, beta distribution) and to illustrate the transition from the preferential release of younger water volumes (examples $\omega_1$ and $\omega_2$) to the random sampling case ($\omega_3$) and the preferential release of older waters ($\omega_4$). The time-variant power-law SAS ($\omega_1$) was obtained by using equation (6) with a time-variant exponent $k(t) = k_{Q1} + [1 - wi(t)](k_{Q2} - k_{Q1})$, with parameters $k_{Q1}$ and $k_{Q2}$ corresponding to the exponent $k$ during the wettest ($wi = 1$) and driest ($wi = 0$) conditions.

Two different examples of solute transport were simulated in the test. In the first case, solute input concentration was generated by adding noise to a sinusoidal wave with annual cycle. This example can be representative of atmospheric solutes





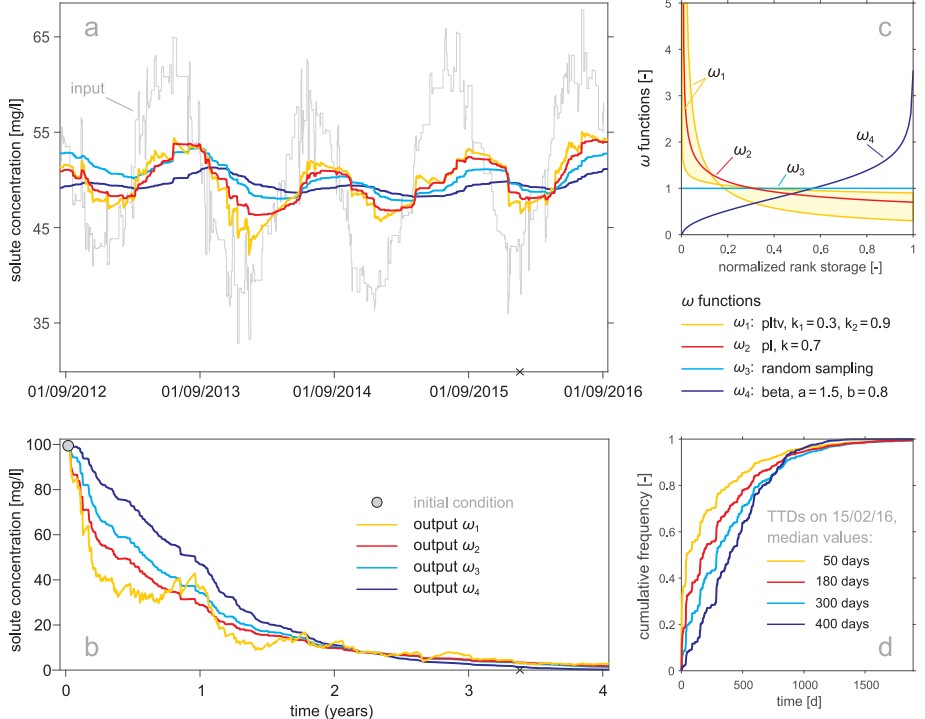

**Figure 3.** Example of results that can be obtained from the model. a) streamflow solute response in case of synusoidal tracer input; b) streamflow solute response in case of step-reduction of the tracer input; c) illustration of the different $\omega_Q$ used in the simulations and listed in Table 1 (as $\omega_1$ is time-variant, its possible shapes are represented by a colored band); d) cumulative travel time distributions (TTDs) extracted on a specific day (15 February 2016, indicated with a cross in plots a) and b)). All simulations share the same settings and only differ in the choice of the $\omega_Q$ function.

with a yearly period. In the second case, the initial storage was set to a concentration of 100 mg/l and any subsequent input was assigned a concentration of 0 mg/l, causing the system to dilute. This example can be representative of a diluting system, e.g. a catchment with agricultural inputs that undergoes a step reduction. Results of both examples are shown in Figure 3.

Each discharge SAS function simulates different transport mechanisms and provides rather different outputs, both in terms
5   of water age and streamflow concentration. In the first solute transport example (Figure 3a), discharge concentration gets progressively damped and shifted as the SAS function moves from younger-water preference to older-water preference. The travel time distributions extracted for February 15th, 2016 (Figure 3d) show that the median age of streamflow may vary by a factor of 3-8 simply based on the selection of the SAS function. The affinity for younger water is rather typical in catchments, at least during wet conditions, while the release of older water is more representative of soil columns or aquifers.
10   The second solute example (Figure 3b) evaluates the "memory" of a system, i.e. the time needed to adapt to a new condition. Again, the preferential release of older storage volumes and the implied lack of young water in streamflow makes the system response more damped. However, this also means that the old water gets depleted faster, hence in the long term (e.g. after 2



years in Figure 3b) the trend is reversed and the residual legacy of the initial conditions is stronger in systems with a high affinity for younger water. The time-variant SAS function ($\omega_1$) is particularly illustrative in this example, because it shows that streamflow concentration can increase in time (e.g. around year 1 in Figure 3b), even in the absence of new solute input, just as a consequence of the changing transport mechanisms. Overall, these quick examples were used to illustrate the model

capabilities, but many other applications to solute transport can be designed and addressed through the model.

## 5    Discussion

### 5.1    Model verification

We evaluate here the numerical accuracy of the model in computing the solution of the age ME (i.e., the rank storage $S_T$) and streamflow concentration $C_Q$. The numerical model is first evaluated by comparing our modified Euler solution (equation 9)

to a numerical implementation of the analytic solution (equation 1). This comparison is only possible for the case of random sampling (RS, section 2.4), as no analytic solution is usually available for other transport schemes. Then, the comparison is made for other shapes of the SAS function, approximating the 'true' solution with a higher-order implementation of equation (8). As in section 4, comparisons are made on the example dataset, using daily average fluxes and the sinusoidal tracer input concentration.

For the RS comparison, the analytic solution was obtained by implementing equation (7) at daily scale, considering that fluxes are piecewise constant while the storage is piecewise linear during the timestep. The numerical solution for the RS was obtained by setting both $\Omega_Q$ and $\Omega_{ET}$ as power laws with parameters $k_Q = k_{ET} = 1$. The numerical model was run for 8 different aggregation timesteps $h$: 1, 2, 3, 4, 6, 8, 12, 24 hours. For each run, the resulting streamflow concentration and one rank storage (corresponding to the end of day 2745) were used for comparison with the analytic solution. Models were run for

8 years using 4 years of spinup. To allow direct comparisons across different aggregation timesteps, streamflow concentrations were extracted at the end of each day, resulting in 8 different timeseries (one per $h$) of 2920 elements. The timeseries were then normalized by the mean and standard deviation of the analytic solution. A timeseries of model errors on streamflow concentration ($err_{C_Q}$) was finally obtained from the difference between the analytic and the numerical (normalized) solutions. The rank storage was evaluated on the entire age domain every 24 hours (again, to allow comparisons across different timesteps).

To avoid comparisons between cumulative functions, the rank storage was used to compute the storage age pdf $p_S$ (see Section 2.1). The errors on $p_S$ were obtained from the difference between the analytic and the numerical solutions. In this case, the error timeseries ($err_{p_S}$) consists, for each of the 8 aggregation timesteps, of 2745 elements. For additional comparisons, the performance of our numerical implementation ("EF*") was compared to the classic implementation of the forward Euler scheme ("EF", i.e., equation (10) with $e[j] = 0$). Results are obtained for 4 different values of the initial storage $S_0$: 300, 500,

1000, 2000 mm. The standard deviations of $err_{p_S}$ and $err_{C_Q}$ are shown in Figure 4 as a function of the aggregation timestep. The EF and EF* implementations almost have the same error on $p_S$, indicating that accounting for the event water does not have a major impact on the overall solution of the age ME. However, as different ages do not contribute equally to streamflow, the event water can have a larger impact on streamflow concentration. This is evident in the performance on $err_{C_Q}$, where the





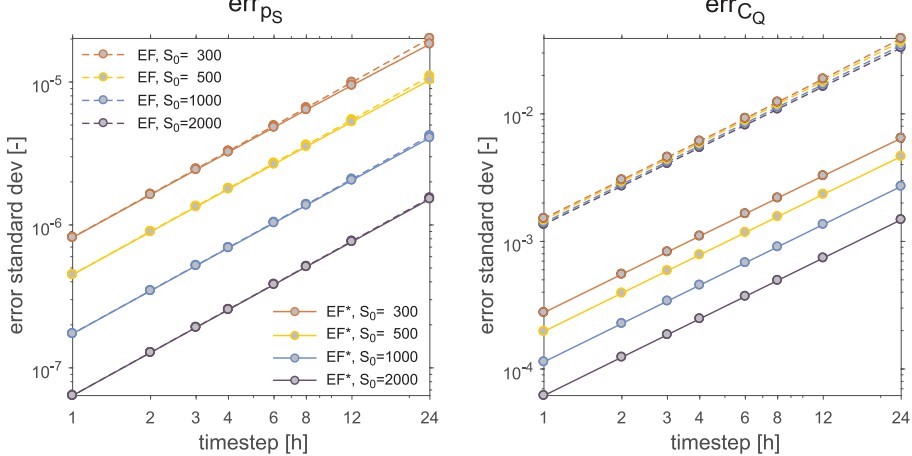

**Figure 4.** Numerical errors on the storage age distribution (left panel) and on streamflow concentration (right panel) as a function of the aggregation timestep. The error timeseries are summarized through their standard deviation. Each plot shows the performance of 2 different numerical schemes: classic Euler Forward (EF) and modified Euler Forward (EF*, which is the default model version). The EF* implementation shows significant improvements with respect to EF in the accuracy of streamflow concentration.

modified EF* implementation is about one order of magnitude more accurate than the classic Euler scheme. The error is on average smaller than $10^{-2}$ the variance of the $C_Q$ signal, which is lower than most measurement errors. The performance on $err_{C_Q}$ also shows that the errors tend to grow with decreasing values of the mean storage, i.e. when the storage gets depleted (or filled) faster. The error of the EF* scheme shows a good stability. This is not surprising as the RS case resembles a linear

reservoir (see Section 2.4) with a coefficient $c$ approximately equal to the mean ratio between the fluxes and the storage $\langle J/S \rangle$ during a timestep. The stability condition for the Euler Forward scheme in the case of a linear reservoir requires that $c < 2/h$ (no fast decay). In typical hydrologic applications, fluxes are usually much smaller than the storage, hence $\langle J/S \rangle \ll 1/h$ and the EF solution is stable.

Results show that the numerical implementation of the ME is satisfactory for the RS solution both in terms of accuracy and

stability. However, solutions other than the RS case may be more challenging owing to the non-uniform age selection played by the outflows. For this reason, we tested power-law SAS functions (equation (6)) with different values of the exponent $k$: 0.2, 0.3, 0.5, 0.7, 1, 1.2, 1.5, 2, 3. The same exponent was used each time for both $\Omega_Q$ and $\Omega_{ET}$. The model was run with a fixed initial storage $S_0 = 1000$, for the same timespan and aggregation timesteps as in the RS case, and the performance was again evaluated in terms of $err_{p_S}$ and $err_{C_Q}$. Given the lack of analytical solutions, we approximated the true solution by

using a higher-order implementation (built-in MATLAB solver 'ode113' (Shampine and Reichelt, 1997)) for equation (8). An example of $C_Q$ timeseries obtained from the different values of $k$ for $h = 24$ hours is reported in Figure 5. The $C_Q$ timeseries are rather different, being progressively more lagged and damped for increasing values of $k$. Although the residual with respect to the higher-order solution can occasionally be up to 1.3 mg/l, it is on average very low compared to the signal, so in this case the accuracy of the model is satisfactory even for $h = 24$ hours. Note that for this dataset, the parameters of the SAS



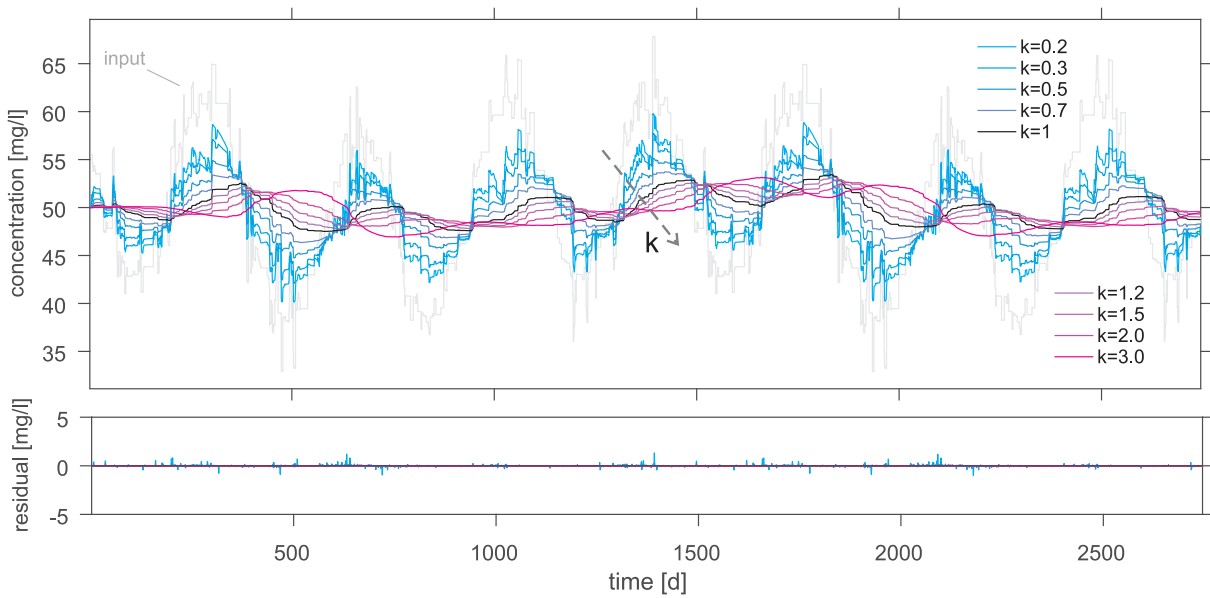

**Figure 5.** Solute concentration ($C_Q$) timeseries obtained from power-law SAS functions with parameter $S_0 = 1000$ and parameter $k \in [0.2, 3.0]$, using a 24-hour timestep (top panel). The timeseries are rather different, being progressively more lagged and damped for increasing values of $k$. The difference with the higher-order solution forms the residual timeseries (bottom panel, same scale as top panel). Residuals are overall limited and they do not cumulate during the 8-year simulation.

function ($k = 0.2$ and $S_0 = 1000$) imply that 30% of the input, on average, becomes output during the same day. The residuals are overall low and do not accumulate during the 8-year simulation, suggesting that even the 24-hour simulation is stable. The performance on $C_Q$ was further evaluated in the same way as for the RS case: we normalized the concentration signals and obtained the error timeseries $err_{C_Q}$ from the difference with the higher-order solution. Similarly, we computed the errors $err_{p_S}$

with respect to the higher-order solution for simulation day 2745. The standard deviations of the errors are shown in Figure 6 for different values of $k$ and aggregation timesteps. The errors on $p_S$ grow for increasing preference of the SAS functions for the younger stored volumes (lower values of $k$). This indicates that the young water preference is a more challenging numerical condition for the solution of the age ME. This behavior is to be mostly attributed to the errors on the youngest waters in storage. Although we use a modified version of the EF scheme to account for the presence of event water in the outflows (equation

(10)), this approximation has some limitations. In particular, the youngest age in storage ($e[j]$) is quantified through the SAS function from previous timestep, so it may give rise to errors at the onset of intense storm events. The interpretation of the behavior of the error on $C_Q$ (Figure 6b) is less straightforward as the errors on the solution $p_S$ can be amplified in various ways by the different SAS functions. Errors appear not too dissimilar for $k$ in the range 0.5-1.2 and they all are reduced by 1 order of magnitude moving from daily to hourly timesteps. The more "extreme" age selections (i.e. $k \leq 0.3$ and $k \geq 2$) tend to result

in higher errors, although the error magnitude remains low (less than $10^{-2}$ the signal variance) and the solution is stable.





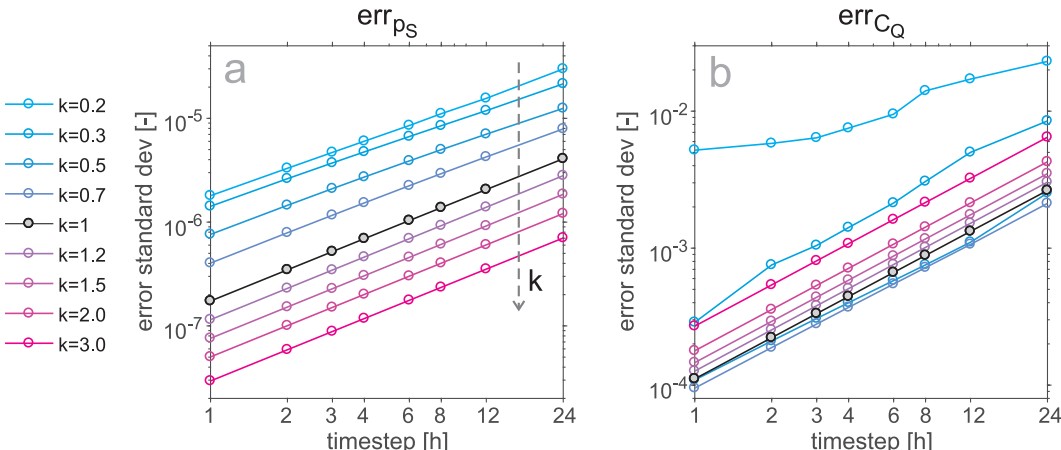

**Figure 6.** Numerical errors on the storage age distribution (a) and on streamflow concentration (b) as a function of the aggregation timestep. The error timeseries are summarized through their standard deviation. Each plot shows the model performance for several shapes of the SAS function, parameterized as a power-law distribution with parameter $k$ (equation (6)). The color code is the same as in Figure 5. The random-sampling case (i.e. $k = 1$) is indicated in black and it is equivalent to the curves featuring $S_0 = 1000$ in Figure 4.

These examples suggest that the behavior of the system can be interpreted using a (non-linear) reservoir analogy. Each individual water parcel can be seen as a depleting reservoir that decreases in time owing to the particular outflow removal (equation 8). This removal is mediated by the SAS functions, so it can become large corresponding to high values of $\omega(S_T, t)$, potentially leading to an unstable fast-decay. The depletion pattern of the reservoir is rather complex as it is nonlinear and it

changes at every timestep, but it suggests that very pronounced age selections should be considered carefully and checked for potential numerical instabilities. Note that for illustration purposes the effects of the two power-law SAS function parameters $k$ and $S_0$ were presented separately (Figures 4 and 6), but they should be considered together as lower storage values may enhance the selection of younger/older waters and increase the numerical errors. The model was here tested for several shapes of the SAS functions on a realistic hydrochemical dataset. Although every dataset is different and it would be impossible to

do a model verification valid for all applications, these results provide some first guidelines as to where the explicit numerical implementation may become critical.

### 5.2 Model applicability, limitations and perspectives

The model is based on a catchment-scale approach, so it only requires catchment-scale fluxes like precipitation, discharge and evapotranspiration. These fluxes can often be measured (or modeled in the case of ET) without the need for a full hydrologic

model. Moreover, the 'pure' SAS function approach implies that, differently from previous approaches (e.g. Bertuzzo et al., 2013; Benettin et al., 2015a), the transport equations which are solved in the model are completely decoupled from the way fluxes were obtained. This notably reduces the number of involved parameters and it simplifies the applicability of the model to different datasets and contexts. Although more research is needed to classify the expected shapes of the SAS functions based





on measurable catchment properties, one can quickly obtain first-order evaluations of solute transport by using SAS functions already tested in the literature (e.g. van der Velde et al., 2015; Harman, 2015; Queloz et al., 2015b; Benettin et al., 2017; Wilusz et al., 2017) and a reasonable choice of the initial storage $S_0$.

The use of an explicit numerical scheme has the potential of greatly reducing the computational times. Short aggregation
timesteps are generally recommended, especially when testing the affinity for younger storage volumes (e.g. equation (6) with parameter $k < 0.3$), but in case larger timesteps (e.g. $h =$24 h) prove satisfactory, the model can typically run in less than a second on a normal computer. The short computational times make the use of calibration techniques easier and the model structure is directly compatible with the DREAM (Vrugt et al., 2009; ter Braak and Vrugt, 2008) calibration packages. The model can be made faster by not considering the zero-precipitation times but, as explained in section 3.3, this improvement is
currently not implemented to keep the model more intuitive.

The model is based on a catchment-scale formulation of transport processes, so it cannot provide spatial information unless the system is partitioned into a series of spatial compartments (e.g. Soulsby et al., 2015). Even in this case, one would need to know the fluxes to/from each compartment, hence losing one of the main advantages of the general SAS approach. The catchment-scale nature of the formulation also implies that SAS functions have a conceptual character and they cannot be
determined directly from physical properties of the system. Their general shape, however, can be traced back to elementary advection-dispersion processes (Benettin et al., 2013) and the mechanistic basis for time-variable SAS functions has recently been highlighted (Pangle et al., 2017).

Although the numerical accuracy of the computations has to be evaluated for each different application, section 5.1 provides some first guidelines to cases where the numerical accuracy may not be satisfactory. Systems whose storage is quickly depleted
by the fluxes are prone to inaccuracies and instabilities. This can happen, for instance, if the system storage is small compared to the fluxes and the SAS functions have a very strong preference for some storage portions. In such cases, higher order schemes may become desirable. The model package already provides a higher-order solution to equation (8) (obtained through the MATLAB built-in function 'ode113'), that can help evaluating the numerical accuracy of the results.

The codes implemented in the *tran*-SAS package can be used to simulate the transport of conservative solutes through a
catchment. This represents a first step towards the modeling of large-scale solute transport. Reactive transport equations can be easily implemented in the main model routine (section 3.2) using effective formulations that integrate biogeochemical processes across the catchment heterogeneity (Rinaldo and Marani, 1987). Being based on a travel time formulation of transport, the model is obviously not suited to simulate the circulation of solutes for which the chronology of the inputs is irrelevant. For a number of cases of interest, however, both the time of entry into the catchment and the residence time of water within
the catchment storage may play an important role in the transport process. Many such examples have been addressed in the literature using a catchment-scale approach, including the case of nitrate export from agricultural catchments (Botter et al., 2006; van der Velde et al., 2012), solutes influenced by evapoconcentration effects (Queloz et al., 2015b), pesticide transport (Bertuzzo et al., 2013; Lutz et al., 2017) and solutes produced by mineral weathering (Benettin et al., 2015a). The provided codes are designed to be easy to understand, so that they can be easily customized by the user and adapted to different contexts





and applications. The next step is then to adapt the model to real-world problems, where solutes' non-conservative behavior has to be taken into account.

## 6  Conclusions

The *tran*-SAS package includes a basic implementation of the age Master Equation (equation 1) using general SAS-functions.

The codes can be used to simulate the transport of solutes through a catchment and to evaluate water residence times. The package is ready-to-go and it includes some example data that can be used to test the main model features. The codes are extensively commented so that they can be edited according to the user's needs. The model is based on a catchment-scale formulation of solute transport and it only relies on measurable data. Main model equations are implemented using an explicit Euler scheme that allows to reduce computational times. The numerical accuracy of the model was verified on the example

data and was shown to be generally satisfactory even at larger (e.g. daily) computation timesteps. The most critical cases are those in which the stored water parcels are rapidly removed by the outflows. This situation can occur when the SAS function assumes very high values for some stored water volumes. In such cases, higher-order model implementations (provided within the package) should be used to check the numerical accuracy of the solution. The model allows to test different SAS functions and evaluate solute transport in the catchment storage and outflows. Applications can be oriented to different catchments and

solutes, advancing our ability to understand and model catchment transport processes.

## 7  Code and Data availability

A maintained code package with example data and documentation is available at the following GitHub repository:
https://github.com/pbenettin/tran-SAS

*Acknowledgements.*  The authors thank Andrea Rinaldo and Gianluca Botter for the useful discussions that inspired this work, and Damiano

Pasetto for support in the numerical implementation of the model equations. PB thanks the ENAC school at EPFL for financial support.



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
