# Peer review of "tran*-SAS v1.0: a numerical model to compute catchment-scale hydrologic *tran*sport using StorAge Selection functions"

_Geoscientific Model Development, 2017_

## Referee Comment (RC1) · Anonymous Referee #1 · 21 Feb 2018

This is a technical paper documenting a model code built on previous developments. This is entirely in keeping with the scope of the journal. A link to open source code is provided (github).

I found this paper exceptionally clearly presented. It's quite easy for the reader to understand what the model does. Illustrative graphics and sensible notational shorthand help in that.

The numerical accuracy test is useful. It seems the authors have taken care of numerical efficiency.

I only had 2 comments related to references to more complex applications:

[Figure]

On p10, l2-3 a catchment with legacy agricultural inputs is put forward as an example of a diluting system such as that simulated here with a synthetic dataset. I don't think that's correct because agricultural inputs are subject to reactive transport, not just dilution, which is not implemented in this code. Please come up with a better example.

On p15, l25-26 the authors claim that reactive transport can be easily implemented. I would question this general statement as especially agricultural solute transport can be quite complex as dissolution, precipitation and re-mobilisation as well as spatial variables (e.g. temperature) matter greatly. Please limit this statement to "simple" reactive transport.

---

## Referee Comment (RC2) · Anonymous Referee #2 · 23 Feb 2018

This study introduces a numerical transport modeling package, *tran*-SAS, which is aimed to simulate solute transport and water residence time at a catchment scale. In particular, the authors explore the computational stability as well as the numerical accuracy of the proposed model. The manuscript is well written, easy to follow, and quite interesting to the hydrologic transport community. A few minor and relatively major points are listed below.

1) Section 1, Line 21: Please revise the first sentence. The new transport model has improved the capabilities in terms of what? I understand your point, but it worth it to make it clear for a general audience. A suggestion is to add two or three sentences on, e.g., how the new transport model can be expressed in different ways depending on the ease of its application in a desired study (Botter et al., GRL2011 vs. van der Velde et al., WRR2012 vs. Harman, WRR2015 vs. Benettin et al., WRR2017). Or, for instance, how this new transport model is much less biased to spatial aggregation as opposed to the traditional approaches assigning the TTD a priori (Danesh-Yazdi et al., GRL2017).

2) Section 2, Line 7: Characterization of the SAS function is also part of the requirements (as emphasized in section 2, line 21) for solving the age distribution of the water storage. Please revise this sentence, accordingly.

3) Equation 3: Isn't this conditional on no precipitation takes place at time $t$? What about those conditions when part of the input precipitation falls directly into the river, contributing to the streamflow? Or what about those conditions when a major portion of the input precipitation contributes rapidly to the streamflow?

4) Section 2, Lines 20-24: $S_0$ and k have been written in different formats in the manuscript (i.e., at one place as bold and italic, and at another place as normal). Please make them consistent throughout the manuscript.

5) Section 2.4, title: I know in their former papers, the authors have already emphasized on the distinction between the random sampling and the well-mixed conditions. As such, I am not sure why they equivalently put them together in this title.

6) Section 3, Line 18: You already called $S_T(0, t_0) = 0$ a "boundary" condition in Eq. (3).

7) Section 3, Line 22: I am not following this last sentence.

8) Page 7, Line 12: Not sure what does 1 in $\Omega_Q[1, j-1]$ imply? It is essentially $S_T(i, j-1)$, so you meant $i$ instead of 1?

9) Page 10, Line 12; Page 11, Lines 1-2: This is an important conclusion, but with a relatively weak reasoning. The difference between the curves in Figure 3b after year 2 is not really

significant. Author might want to provide another example that clearly demonstrates this conclusion.

10) Does the *tran*-SAS package also include the Markov Chain Monte Carlo calibration scheme (with reference to Page 15, Line 8)? If yes, please add a few lines on how such a scheme is embedded within the package. If no, why not to include?

11) The examples include two different ways of parameterizing the SAS functions, that is, using the power-law and the gamma functions. However, there is no discussion about which model provides a better solution to TTD and $C_Q$. This is a missing, but quite important information for the users of this model and should be well addressed in the manuscript.

12) Section 1, Line 22: "such as" instead of "like"? Also, at Page 14, Line 13.

13) Section 2, Line 24: "expressed in terms of" instead of "expressed as"?

14) Section 2: You might want to define $C_Q(t)$ as well to complete your definitions here.

15) Page 11, Line 11: Define the acronym for the random sampling, i.e., RS, earlier in the manuscript where it was first mentioned.

---

## Short Comment (SC1) · 15 Mar 2018

It is GMD policy that authors neeed to provide a persistent access to the exact version of the source code used for the model version presented in the paper. As explained in https://www.geoscientific-model-development.net/about/manuscript_types.html the preferred reference to this release is through the use of a DOI which is then cited in the paper. For projects in GitHub a DOI for a released code version can easily be created using Zenodo, see https://guides.github.com/activities/citable-code/ for details. Please note that in the Code Availability section you can still point the reader to the GitHub repository for the newest version even if you use a DOI for the relevant release.

[Figure]

Lutz Gross GMD Executive Editor

---

## Author Comment (AC1) · 15 Mar 2018

We thank referee 1 for his/her positive assessment of the paper. We agree with the useful comments on reactive transport:

> *On p10, l2-3 a catchment with legacy agricultural inputs is put forward as an example of a diluting system such as that simulated here with a synthetic dataset. I don't think that's correct because agricultural inputs are subject to reactive transport, not just dilution, which is not implemented in this code. Please come up with a better example.*

*On p15, l25-26 the authors claim that reactive transport can be easily implemented. I would question this general statement as especially agricultural solute transport can be quite complex as dissolution, precipitation and re-mobilisation as well as spatial variables (e.g. temperature) matter greatly. Please limit this statement to "simple" reactive transport.*

Indeed, the transport of agricultural solutes (particularly nitrate and phosphorus) is far more complex than a simple dilution and we do not want to convey the idea that a conservative transport model is appropriate in those situations. Agricultural inputs were mentioned just as an example of input that, due to regulation, can undergo rather drastic reductions. We will be more specific, mentioning more conservative solutes like chloride, that can have an agricultural origin (see e.g. van der Velde et al., 2010 and Martin et al., 2004) and a substantially simpler biogeochemical cycling.

REFERENCES:

van der Velde, Y., de Rooij, G. H., Rozemeijer, J. C., van Geer, F. C., Broers, H. P. (2010). Nitrate response of a lowland catchment: On the relation between stream concentration and travel time distribution dynamics. Water Resources Research, 46(11). https://doi.org/10.1029/2010WR009105

Martin, C., Aquilina, L., Gascuel-Odoux, C., Molénat, J., Faucheux, M. and Ruiz, L. (2004), Seasonal and interannual variations of nitrate and chloride in stream waters related to spatial and temporal patterns of groundwater concentrations in agricultural catchments. Hydrol. Process., 18: 1237–1254. https://doi.org/10.1002/hyp.1395

---

## Author Comment (AC2) · 15 Mar 2018

We thank reviewer 2 for the positive comments. The list of points includes insightful suggestions that we will implement in the revised version:

*1) Section 1, Line 21: Please revise the first sentence. The new transport model has improved the capabilities in terms of what? I understand your point, but it worth it to make it clear for a general audience. A suggestion is to add two or three sentences on, e.g., how the new transport model can be expressed in different ways depending on the ease of its application in a desired study (Botter et al., GRL2011 vs. van der Velde et al., WRR2012 vs. Harman, WRR2015*

*vs. Benettin et al., WRR2017). Or, for instance, how this new transport model is much less biased to spatial aggregation as opposed to the traditional approaches assigning the TTD a priori (Danesh-Yazdi et al., GRL2017).*

We agree and we will add a paragraph to explain the advantages of the formulation of transport by SAS-functions.

*2) Section 2, Line 7: Characterization of the SAS function is also part of the requirements (as emphasized in section 2, line 21) for solving the age distribution of the water storage. Please revise this sentence, accordingly.*

We will review this sentence to further clarify this crucial point. Thanks for the suggestion.

*3) Equation 3: Isn't this conditional on no precipitation takes place at time t? What about those conditions when part of the input precipitation falls directly into the river, contributing to the streamflow? Or what about those conditions when a major portion of the input precipitation contributes rapidly to the streamflow?*

This boundary condition is coupled to equation (1) where precipitation $J(t)$ appears explicitly. For any age $\varepsilon > 0$ (even very small), integration of eq (1) results in $S_T(\varepsilon, t) > 0$ (if precipitation occurred during $\varepsilon$).

*4) Section 2, Lines 20-24: S0 and k have been written in different formats in the manuscript (i.e., at one place as bold and italic, and at another place as normal). Please make them consistent throughout the manuscript.*

We will correct this. Thanks for pointing this out.

*5) Section 2.4, title: I know in their former papers, the authors have already emphasized on the distinction between the random sampling and the well-mixed conditions. As such, I am not sure why they equivalently put them together in this title.*

Indeed, there are differences between a "well-mixed" and a "randomly-sampled" reservoir and we do not imply that the two coincide. In a catchment-scale formulation this distinction has no practical consequences and equation (7) holds in both cases. As readers are usually more familiar with the concept of "well-mixed", we prefer to keep the title as is.

*6) Section 3, Line 18: You already called ST (0, t0) = 0 a "boundary" condition in Eq. (3).*

This is true, but equation (8) is now an ordinary differential equation in the variable $T$, so the condition $T = 0$ is an initial condition.

*7) Section 3, Line 22: I am not following this last sentence.*

We will reformulate it to make it clearer. The variable $S_T(T, T + t_0)$ describes the amount of water storage that is younger than the water entered in $t_0$. In other words, it describes the amount of water entered after $t_0$ that is still inside the system. When time grows, water entered after $t_0$ gradually replaces the water entered before $t_0$, and for very large $T$ all the water storage is made of water entered after $t_0$.

*8) Page 7, Line 12: Not sure what does 1 in $\Omega Q[1, j − 1]$ imply? It is essentially ST (i, j − 1), so you meant i instead of 1?*

$e[j]$ represents an estimate of the event water so it refers to the first element in the rank storage $S_T(i = 1, j − 1)$.

*9) Page 10, Line 12; Page 11, Lines 1-2: This is an important conclusion, but with a relatively weak reasoning. The difference between the curves in Figure 3b after year 2 is not really 2 significant. Author might want to provide another example that clearly demonstrates this conclusion.*

We agree with this comment. The conclusion is more of a general consequence of the young storage preference, but it is not well visible in Figure 3b. We will either modify Figure 3 or suggest other examples from the literature where this was more evident.

*10) Does the tran-SAS package also include the Markov Chain Monte Carlo calibration scheme (with reference to Page 15, Line 8)? If yes, please add a few lines on how such a scheme is embedded within the package. If no, why not to include?*

The MCMC package could not be included in our package for copyright reasons, but the structure of the model function is fully compatible with the DREAM ZS [ter Braak and Vrugt, 2008, Vrugt et al., 2009] software for matlab freely available at http://faculty.sites.uci.edu/jasper/software/.

*11) The examples include two different ways of parameterizing the SAS functions, that is, using the power-law and the gamma functions. However, there is no discussion about which model provides a better solution to TTD and CQ. This is a missing, but quite important information for the users of this model and should be well addressed in the manuscript.*

The examples actually included parametrization using power-law or beta functions. It is not possible to tell apriori which function provides a better solution because it depends on the specific application and on the desired degree of complexity of the model. For example, the beta function is a more general case than the power-law, so in principle it provides more accurate solutions, but it also makes use of more parameters and it involves longer computations. We will better highlight this point.

*12) Section 1, Line 22: "such as" instead of "like"? Also, at Page 14, Line 13.*
*13) Section 2, Line 24: "expressed in terms of" instead of "expressed as"?*
*14) Section 2: You might want to define CQ(t) as well to complete your definitions here.*
*15) Page 11, Line 11: Define the acronym for the random sampling, i.e., RS, earlier in the manuscript where it was first mentioned.*

We thank reviewer 2 for these suggestions. We will modify the manuscript accordingly.

[Figure]

REFERENCES

ter Braak, C. J. F., Vrugt, J. a. (2008). Differential Evolution Markov Chain with snooker updater and fewer chains. Statistics and Computing, 18(4), 435–446. https://doi.org/10.1007/s11222-008-9104-9

Vrugt, J. A., ter Braak, C. J. F., Diks, C., Robinson, B. A., Hyman, J. M., Higdon, D. (2009). Accelerating Markov Chain Monte Carlo Simulation by Differential Evolution with Self-Adaptive Randomized Subspace Sampling. International Journal of Nonlinear Sciences and Numerical Simulation, 10(3), 271–288. https://doi.org/10.1515/IJNSNS.2009.10.3.273

---

## Author Comment (AC3) · 15 Mar 2018

We thank dr. Gross for this very useful comment. We had not realized that Zenodo DOI's would support software versioning, so we had decided to provide the present package version as supplementary material and then create a DOI after the manuscript revision (in case changes to the code were necessary). We will now link the GitHub project to Zenodo, so it will automatically generate a DOI at each software release.

---

## Referee Comment (RC3) · Anonymous Referee #3 · 19 Mar 2018

The authors present a very useful Matlab implementation of the StorAge Selection modeling framework.

The implementation is essentially a solute transport model, because the solute concentrations are one of the state variables. With the SAS framework it is possible to calculate solute concentrations "offline", by storing the travel time distribution of stream flow for select (sampled) times, and multiplying these with the tracer input history. This is efficient for a small number of samples and a large number of tracers. Perhaps not a common case.

From the manuscript, it is not clear if the model supports simultaneous calculation of

multiple solutes. Perhaps I missed that. It would be useful.

I would like to see a stronger encouragement by the authors to test the parameter space for each new case. The example parameters are very hypothetical.

The model description is accurate and easy to understand (for someone who has worked with a different implementation of a SAS model). I hope one of the other reviewers is a "SAS dummy" who can ask the questions that seem obvious to me.

I have a few comments specific to the text:

P5 L19, Eq 6: This implementation is equivalent to the fractional StorAge Selection (fSAS) implementation, right?

Section 3.3: I would like the authors to elaborate on the discussion of the "old pool". Transient tracers like tritium and chlorine-36 demand that the age distribution of the old pool is accurately represented. Or at least in the concentration in the "old pool" needs to be represented.

P9 L6: "long term" = 4 years? P9 L13: S0=1000? mm? P9 L11: I understand the parameterization of the example is not intended to represent the hydrogeological conditions of the particular data set. Nevertheless, I find the random sample (kET=1) surprising, as I would expect the vegetation to have even the slightest preference for younger water. Perhaps the authors can warn the reader that these parameters should not be considered "valid" for any catchment and encourage the user of the tranSAS to vary all parameters of the example case drastically if applied to a specific setting to test the sensitivity.

Figure 3d, please clarify that this is the stream flow TTD.

P10 L1: "solutes with a yearly period".... like stable isotopes of water? (These aren't really solutes.)

P10 L8: The range in median ages can vary much more. It all depends on the fictional

parameters you enter into your model. It might be more relevant to compare the non-random-sampling cases with the random-sampling case. Or reiterate that any power with a k < 1 prefers younger water and will therefore have a younger TTDs (right? or is this not alwyas the case?)

P10 L10: This dilution example is interesting. Is it true that the stream solute concentration is the inverse of the TTD in the random sampling case (k=1)? It might be worth mentioning. The inverse problem, a step increase of a contaminant input relates more directly to the TTD. I do like this example because it is more optimistic about the potential to reduce environmental contamination. And it illustrates an important aspect of transient contaminant flow, that even with zero input, stream concentrations can increase due to the variable hydrology.

P15 L6: "less than a second" for a 4 year time series? How much longer does the ode113 solution take?

P15 L28: "chronology of the inputs is irrelevant" Not quite sure how to interpret this. The chronology of a constant input decaying tracer (e.g. tritium for the last 30 years) is irrelevant, in the sense that it doesn't matter "when" the precipitation entered the catchment, but it does matter "how long ago". I know what is menat, but it reads like this model is only relevant for tracers with input fluctuations, which isn't the case (as long as the tracer decays on relevant time scales).

---

## Author Comment (AC4) · 19 Mar 2018

We thank Referee 3 for his/her positive evaluation of the model

> *The authors present a very useful Matlab implementation of the StorAge Selection modeling framework.*
> *The implementation is essentially a solute transport model, because the solute concentrations are one of the state variables. With the SAS framework it is possible to calculate solute concentrations "offline", by storing the travel time distribution of stream flow for select (sampled) times, and multiplying these with the tracer input history. This is efficient for a small number of samples and a large*

[Figure]

*number of tracers. Perhaps not a common case.*

*From the manuscript, it is not clear if the model supports simultaneous calcula-tion of multiple solutes. Perhaps I missed that. It would be useful.*

The present implementation of the SAS framework is chiefly oriented to modeling the time-evolution of one solute in a hydrologic system. Extending the code to the case of multiple solutes is an easy task because the water carrier (and its transit time distribu-tions) remains the same. Hence, one only needs to duplicate the equations that involve solute transport (or re-run the code with modified initial and boundary conditions). We prefer to keep this basic code simple and intuitive, and let the user adapt the code to more advanced transport problems.

*I would like to see a stronger encouragement by the authors to test the parameter space for each new case. The example parameters are very hypothetical.*

We agree with Referee 3 that the parameter space should be widely explored and we will highlight this point in the revised version.

*The model description is accurate and easy to understand (for someone who has worked with a different implementation of a SAS model). I hope one of the other reviewers is a "SAS dummy" who can ask the questions that seem obvious to me.*

We thank Referee 3 as, indeed, we put quite some effort to make the code description and implementation easy to understand.

*I have a few comments specific to the text: P5 L19, Eq 6: This implementation is equivalent to the fractional StorAge Selection (fSAS) implementation, right?*

Mathematically, equation (6) becomes a fSAS after the variable transformation $S_T(T,t) \rightarrow f(T,t) = S_T(T,t)/S(t)$.

*Section 3.3: I would like the authors to elaborate on the discussion of the "old pool". Transient tracers like tritium and chlorine-36 demand that the age distribu-*

*tion of the old pool is accurately represented. Or at least in the concentration in the "old pool" needs to be represented.*

We agree with Referee 3. The problem of what is to be considered as "old" also depends on the considered tracer and its characteristic input timescales. In the case of tracers like tritium and chlorine-36, a much longer spin-up is advised to limit the impact of apriori assumptions on the initial old pool concentration. Also in this case a much longer timestep (e.g. weeks) could be used in the computations.

*P9 L6: "long term" = 4 years? P9 L13: S0=1000? mm? P9 L11: I understand the parameterization of the example is not intended to represent the hydrogeological conditions of the particular data set. Nevertheless, I find the random sample (kET=1) surprising, as I would expect the vegetation to have even the slightest preference for younger water. Perhaps the authors can warn the reader that these parameters should not be considered "valid" for any catchment and encourage the user of the tranSAS to vary all parameters of the example case drastically if applied to a specific setting to test the sensitivity.*

As mentioned in previous comments, we fully agree with Referee 3 on this point. These were just hypothetical parameters (although they are similar to parameters found in small catchments in wet climates, e.g. Benettin et al., 2017) and should not be taken as representative of a general catchment behavior. We will clarify this in the revised manuscript. We also believe that, thanks to the short computational times, the tranSAS code facilitates sensitivity analyses.

*Figure 3d, please clarify that this is the stream flow TTD.*
*P10 L1: "solutes with a yearly period".... like stable isotopes of water? (These aren't really solutes.)*

We will correct this, thanks for pointing it out.

*P10 L8: The range in median ages can vary much more. It all depends on*

*the fictional parameters you enter into your model. It might be more relevant to compare the nonrandom-sampling cases with the random-sampling case. Or reiterate that any power with a k < 1 prefers younger water and will therefore have a younger TTDs (right? or is this not alwyas the case?)*

Age estimates are typically more sensitive to model parameters than solute concentration estimates. We will specify this by expanding the discussion on the sensitivity of model results. The relationship between the age distribution and the value of parameter k is not straightforward as it also depends on which portion of the age distribution is considered. We will modify this paragraph to highlight the differences with respect to the random-sampling case.

*P10 L10: This dilution example is interesting. Is it true that the stream solute concentration is the inverse of the TTD in the random sampling case (k=1)? It might be worth mentioning. The inverse problem, a step increase of a contaminant input relates more directly to the TTD. I do like this example because it is more optimistic about the potential to reduce environmental contamination. And it illustrates an important aspect of transient contaminant flow, that even with zero input, stream concentrations can increase due to the variable hydrology.*

We thank Referee 3 for this positive comment.

*P15 L6: "less than a second" for a 4 year time series? How much longer does the ode113 solution take?*

On an ordinary PC, the test-case implementation (4 years spin-up + 4 years run, power-law SAS functions with k=0.7, 24-hour timestep) runs in less than a second for the modified Euler Scheme and in about 30 seconds for the ode113 solution.

*P15 L28: "chronology of the inputs is irrelevant" Not quite sure how to interpret this. The chronology of a constant input decaying tracer (e.g. tritium for the last 30 years) is irrelevant, in the sense that it doesn't matter "when" the precipitation*

*entered the catchment, but it does matter "how long ago". I know what is menat, but it reads like this model is only relevant for tracers with input fluctuations, which isn't the case (as long as the tracer decays on relevant time scales).*

In our view, the impact of input "chronology" is twofold: it expresses the time-variability of the input and it also determines the residence time of the input in the system (traditionally seen as the interval between present time and entrance time). In this paragraph we wanted to warn the reader that sometimes solute concentration can be driven by factors that do not depend on when the input entered the system nor on how long it remained in the system. We will clarify this point.
* * *

---

## Author Response (AR1)

**manuscript gmd-2017-305**

tran-SAS v1.0: a numerical model to compute catchment-scale hydrologic transport using StorAge Selection functions

Dear Editor,

we have modified the paper according to the comments provided by the referees. As the numerical model did not need any major change, we have not created a new model release. Following the suggestion by Dr. Gross, the current release (tran-SAS v1.0) has been assigned a zenodo DOI (https://doi.org/10.5281/zenodo.1203600) and this has been included in the code availability section. Overall, we are grateful to the three referees and to Dr. Gross for their constructive comments which helped improve our manuscript.

Yours sincerely,

Paolo Benettin and Enrico Bertuzzo

**RESPONSE TO REFEREES' COMMENTS**

In the following, referees' comments are reported in italic. Our responses follow point-by-point

**REVIEWER 1**

*This is a technical paper documenting a model code built on previous developments. This is entirely in keeping with the scope of the journal. A link to open source code is provided (github). I found this paper exceptionally clearly presented. It's quite easy for the reader to understand what the model does. Illustrative graphics and sensible notational shorthand help in that. The numerical accuracy test is useful. It seems the authors have taken care of numerical efficiency.*

We thank Referee 1 for her/his very positive assessment of the paper.

*I only had 2 comments related to references to more complex applications:*

*"On p10, l2-3 a catchment with legacy agricultural inputs is put forward as an example of a diluting system such as that simulated here with a synthetic dataset. I don't think that's correct because agricultural inputs are subject to reactive transport, not just dilution, which is not implemented in this code. Please come up with a better example."*

*"On p15, l25-26 the authors claim that reactive transport can be easily implemented. I would question this general statement as especially agricultural solute transport can be quite complex as dissolution, precipitation and re-mobilisation as well as spatial variables (e.g. temperature) matter greatly. Please limit this statement to "simple" reactive transport."*

We agree with these useful comments on reactive transport. Indeed, the transport of agricultural solutes (particularly nitrate and phosphorus) is far more complex than a simple dilution and we do not want to convey the idea that a conservative transport model is appropriate in those situations. Agricultural inputs were mentioned just as an example of input that, due to regulation, can undergo rather drastic reductions. We now refer explicitly to the case of chloride (Page 10 Lines 1-2), that can have an agricultural origin (see e.g. van der Velde et al., 2010 and Martin et al., 2004) and a substantially simpler biogeochemical cycling.

**REVIEWER 2**

*This study introduces a numerical transport modeling package, tran-SAS, which is aimed to simulate solute transport and water residence time at a catchment scale. In particular, the authors explore the computational stability as well as the numerical accuracy of the proposed model. The manuscript is well written, easy to follow, and quite interesting to the hydrologic transport community. A few minor and relatively major points are listed below.*

We thank Reviewer 2 for the positive comments.

*1) Section 1, Line 21: Please revise the first sentence. The new transport model has improved the capabilities in terms of what? I understand your point, but it worth it to make it clear for a general audience. A suggestion is to add two or three sentences on, e.g., how the new transport model can be expressed in different ways depending on the ease of its application in a desired study (Botter et al., GRL2011 vs. van der Velde et al., WRR2012 vs. Harman, WRR2015 vs. Benettin et al., WRR2017). Or, for instance, how this new transport model is much less biased to spatial aggregation as opposed to the traditional approaches assigning the TTD a priori (Danesh-Yazdi et al., GRL2017).*

We think that an exhaustive discussion on the differences between SAS function parametrization methods already exists in the literature (see e.g. Harman, 2015), so we have modified our introduction to accommodate this Reviewer's suggestion, but limited to the comment that the SAS approach is less biased to spatial aggregation (P2 L22).

*2) Section 2, Line 7: Characterization of the SAS function is also part of the requirements (as emphasized in section 2, line 21) for solving the age distribution of the water storage. Please revise this sentence, accordingly.*

We have now specified all the requirements for the application of the approach at the beginning of Section 4 (Application Example).

*3) Equation 3: Isn't this conditional on no precipitation takes place at time t? What about those conditions when part of the input precipitation falls directly into the river, contributing to the streamflow? Or what about those conditions when a major portion of the input precipitation contributes rapidly to the streamflow?*

This boundary condition is coupled to equation (1) where precipitation $J(t)$ appears explicitly. For any age $\varepsilon > 0$ (even very small), integration of eq (1) results in $S_T(\varepsilon, t) > 0$ (if precipitation occurred during $\varepsilon$).

*4) Section 2, Lines 20-24: S0 and k have been written in different formats in the manuscript (i.e., at one place as bold and italic, and at another place as normal). Please make them consistent throughout the manuscript.*

We decided to remove the bold format and only use italic in the formulas. Thanks for pointing this out.

*5) Section 2.4, title: I know in their former papers, the authors have already emphasized on the distinction between the random sampling and the well-mixed conditions. As such, I am not sure why they equivalently put them together in this title.*

Indeed, there are differences between a "well-mixed" and a "randomly-sampled" reservoir and we do not imply that the two coincide. In a catchment-scale formulation this distinction has no practical consequences and equation (7) holds in both cases. As readers are usually more familiar with the concept of "well-mixed", we preferred to keep the section title as is.

*6) Section 3, Line 18: You already called ST (0, t0) = 0 a "boundary" condition in Eq. (3).*

This is true, but equation (8) is now an ordinary differential equation in the variable T, so the condition T=0 is formally an initial condition.

*7) Section 3, Line 22: I am not following this last sentence.*

The entire sentence (P6 L17-23) has been reformulated and in particular:
"Water entered after t_0 gradually replaces the water entered before t_0 and for very large T the solution reaches (asymptotically) the total storage in the system, as no water that had entered before t_0 is still present in the system."

> *8) Page 7, Line 12: Not sure what does 1 in ΩQ[1, j − 1] imply? It is essentially ST (i, j − 1), so you meant i instead of 1?*

The term e[j] represents an estimate of the event water so it refers to the first element in the rank storage. As the indexes i and j go from 0 to N, we have now indicated the first element of the rank storage as ST[i = 0, j-1].

> *9) Page 10, Line 12; Page 11, Lines 1-2: This is an important conclusion, but with a relatively weak reasoning. The difference between the curves in Figure 3b after year 2 is not really 2 significant. Author might want to provide another example that clearly demonstrates this conclusion.*

We agree with this comment. The conclusion is more of a general consequence of the young storage preference, but it is not well visible in Figure 3b. We have specified this (P10 L13-14) in the revised text.

> *10) Does the tran-SAS package also include the Markov Chain Monte Carlo calibration scheme (with reference to Page 15, Line 8)? If yes, please add a few lines on how such a scheme is embedded within the package. If no, why not to include?*

The MCMC package could not be included in our package for copyright reasons, but the structure of the model function is fully compatible with the DREAM_ZS [ter Braak and Vrugt, 2008, Vrugt et al., 2009] software for matlab, freely available at http://faculty.sites.uci.edu/jasper/software/.

> *11) The examples include two different ways of parameterizing the SAS functions, that is, using the power-law and the gamma functions. However, there is no discussion about which model provides a better solution to TTD and CQ. This is a missing, but quite important information for the users of this model and should be well addressed in the manuscript.*

The examples actually included parametrization using power-law or beta functions. It is not possible to tell *apriori* which function provides a better solution because it depends on the specific application and on the desired degree of complexity of the model. For example, the beta function is a more general case than the power-law, so in principle it provides more accurate solutions, but it also makes use of more parameters and it involves longer computations.

> *12) Section 1, Line 22: "such as" instead of "like"? Also, at Page 14, Line 13.*

> *13) Section 2, Line 24: "expressed in terms of" instead of "expressed as"?*

> *14) Section 2: You might want to define CQ(t) as well to complete your definitions here.*

> *15) Page 11, Line 11: Define the acronym for the random sampling, i.e., RS, earlier in the manuscript where it was first mentioned.*

Done, we thank Reviewer 2 for these corrections.

**REVIEWER 3**

*The authors present a very useful Matlab implementation of the StorAge Selection modeling framework.*

We thank Referee 3 for her/his positive evaluation of the model.

*The implementation is essentially a solute transport model, because the solute concentrations are one of the state variables. With the SAS framework it is possible to calculate solute concentrations "offline", by storing the travel time distribution of stream flow for select (sampled) times, and multiplying these with the tracer input history. This is efficient for a small number of samples and a large number of tracers. Perhaps not a common case.*

*From the manuscript, it is not clear if the model supports simultaneous calculation of multiple solutes. Perhaps I missed that. It would be useful.*

The present implementation of the SAS framework is chiefly oriented to modeling the time-evolution of one solute in a hydrologic system. Extending the code to the case of multiple solutes is an easy task because the water carrier (and its transit time distributions) remains the same. Hence, one only needs to duplicate the equations that involve solute transport (or re-run the code with modified initial and boundary conditions). We preferred to keep this basic code simple and intuitive, and let the user adapt the model to more advanced transport problems.

*I would like to see a stronger encouragement by the authors to test the parameter space for each new case. The example parameters are very hypothetical.*

We agree with Referee 3 that the parameter space should be widely explored and this has been stressed in the revised version (P9 L23-24, P10 L17-22 and P11 L1-2).

*The model description is accurate and easy to understand (for someone who has worked with a different implementation of a SAS model). I hope one of the other reviewers is a "SAS dummy" who can ask the questions that seem obvious to me.*

We thank Referee 3 for this positive comment as, indeed, we put quite some effort to make the code description and implementation easy to understand.

*I have a few comments specific to the text:*

*P5 L19, Eq 6: This implementation is equivalent to the fractional StorAge Selection (fSAS) implementation, right?*

Mathematically, equation (6) becomes a fSAS after the variable transformation $S_T(T,t) \rightarrow f(T,t) = S_T(T,t)/S(t)$.

*Section 3.3: I would like the authors to elaborate on the discussion of the "old pool". Transient tracers like tritium and chlorine-36 demand that the age distribution of the old pool is accurately represented. Or at least in the concentration in the "old pool" needs to be represented.*

We agree with Referee 3. The problem of what is to be considered as "old" also depends on the considered tracer and its characteristic input timescales (see Benettin *et al.*, 2017a for a discussion on this). In the case of tracers like tritium and chlorine-36, a much longer spin-up is advised to limit the impact of *apriori*

assumptions on the initial old pool concentration. Also in this case a much longer timestep (e.g. weeks) could be used in the computations. We have expanded the discussion on this at P8 L17-19 and P9 L2-3.

> *P9 L6: "long term" = 4 years? P9 L13: S0=1000? mm? P9 L11: I understand the parameterization of the example is not intended to represent the hydrogeological conditions of the particular data set. Nevertheless, I find the random sample (kET=1) surprising, as I would expect the vegetation to have even the slightest preference for younger water. Perhaps the authors can warn the reader that these parameters should not be considered "valid" for any catchment and encourage the user of the tranSAS to vary all parameters of the example case drastically if applied to a specific setting to test the sensitivity.*

As mentioned in previous comments, we fully agree with Referee 3 on this point. These were just hypothetical parameters (although they are similar to parameters found in small catchments in wet climates, e.g. Benettin et al., 2017b) and should not be taken as representative of a general catchment behavior. This has been clarified in the revised manuscript (P9 L23-24, P10 L17-22 and P11 L1-2). We also believe that, thanks to the short computational times, the *tran*SAS code facilitates sensitivity analyses.

> *Figure 3d, please clarify that this is the stream flow TTD.*

> *P10 L1: "solutes with a yearly period".... like stable isotopes of water? (These aren't really solutes.)*

Done, thank you.

> *P10 L8: The range in median ages can vary much more. It all depends on the fictional parameters you enter into your model. It might be more relevant to compare the nonrandom-sampling cases with the random-sampling case. Or reiterate that any power with a k < 1 prefers younger water and will therefore have a younger TTDs (right? or is this not alwyas the case?)*

Age estimates are typically more sensitive to model parameters than solute concentration estimates. This has been specified within the expanded discussion on the sensitivity of model results (P10 L17-22 and P11 L1-2). The relationship between the age distribution and the value of parameter k is not straightforward as it also depends on which portion of the age distribution is considered.

> *P10 L10: This dilution example is interesting. Is it true that the stream solute concentration is the inverse of the TTD in the random sampling case (k=1)? It might be worth mentioning.*

We did not fully understand this comment.

> *The inverse problem, a step increase of a contaminant input relates more directly to the TTD. I do like this example because it is more optimistic about the potential to reduce environmental contamination. And it illustrates an important aspect of transient contaminant flow, that even with zero input, stream concentrations can increase due to the variable hydrology.*

We thank Referee 3 for this feedback.

> *P15 L6: "less than a second" for a 4 year time series? How much longer does the ode113 solution take?*

On an ordinary PC, the test-case implementation (4 years spin-up + 4 years run, power-law SAS functions with k=0.7, 24-hour timestep) runs in less than a second for the modified Euler Scheme and in about 30 seconds for the ode113 solution.

> *P15 L28: "chronology of the inputs is irrelevant" Not quite sure how to interpret this. The chronology of a constant input decaying tracer (e.g. tritium for the last 30 years) is irrelevant, in the sense that it doesn't matter "when" the precipitation entered the catchment, but it does matter "how long ago". I know what is menat, but it reads like this model is only relevant for tracers with input fluctuations, which isn't the case (as long as the tracer decays on relevant time scales).*

In our view, the impact of input "chronology" is twofold: it expresses the time-variability of the input and it also determines the residence time of the input in the system (traditionally seen as the interval between present time and entrance time). In this paragraph we wanted to warn the reader that sometimes solute concentration can be driven by factors that do not depend on when the input entered the system nor on how long it remained in the system. We have better specified this second point (P16 L22).

[revised manuscript text omitted]